# Drive-by Methodologies Applied to Railway Infrastructure Subsystems: A Literature Review—Part II: Track and Vehicle

**Cássio Bragança [1,\*]**, **Edson F. Souza [1,2]**, **Diogo Ribeiro [3]**, **Andreia Meixedo [4]**, **Túlio N. Bittencourt [1]** and **Hermes Carvalho [1,5]**

1   Department of Structural and Geotechnical Engineering, University of São Paulo, São Paulo 05508-900, Brazil; eflorentino@usp.br (E.F.S.); tbitten@usp.br (T.N.B.); hermes@dees.ufmg.br (H.C.)
2   Department of Civil Engineering, Federal University of Technology, Guarapuava 85053-525, Brazil
3   CONSTRUCT-LESE, School of Engineering, Polytechnic of Porto, 4200-072 Porto, Portugal; drr@isep.ipp.pt
4   CONSTRUCT-LESE, Faculty of Engineering, University of Porto, 4200-072 Porto, Portugal; ameixedo@fe.up.pt
5   Department of Structural Engineering, Federal University of Minas Gerais, Belo Horizonte 31270-901, Brazil
\*   Correspondence: cassioscb@usp.br

**Abstract:** Railways are one of the most important means of transportation, allowing people and goods to move quickly, environmentally beneficially, and efficiently over long distances. To ensure safe and reliable operations, regular condition-based assessments of trains and track are of paramount importance. Drive-by methodologies, which utilize data collected by onboard monitoring systems as the vehicle travels over the track, have gained popularity as an economically viable strategy for monitoring extensive track networks as well as vehicles traveling on them. This paper presents a critical review of these methodologies applied to railway tracks and vehicles. It assesses research on track irregularities, rail conditions, and the condition of rail supporting elements, highlighting important early developments and recent papers that provide insights into future practical applications. Additionally, the paper explores works related to global vehicle condition evaluation, focusing on the identification of suspension and wheelset element damage and also discussing challenges towards commercial application. The findings suggest that drive-by methodologies have several promising future applications. These include track maintenance optimization, proactive fault detection, predictive maintenance, track performance evaluation, vehicle health monitoring, and data-driven decision-making. By leveraging drive-by assessments, railway operators can optimize maintenance efforts, detect faults early, predict remaining component life, evaluate track performance, monitor vehicle health, and make informed decisions based on data analysis. Finally, a comprehensive conclusion summarizes the achievements thus far and provides perspectives for forthcoming developments. The future practical applications of drive-by methodologies discussed in this review have the potential to revolutionize railway track and vehicle assessments, leading to safer and more efficient railway operations in the days to come.

**Keywords:** drive-by; indirect monitoring; railway bridges; railway tracks; railway vehicles; damage detection

## 1. Introduction

Given their low energy consumption compared to other means of land transport, railways are gaining more and more importance in a world that urges both sustainability and economic growth [1]. This advancement of railways has resulted in an intense technological effort towards increasing both speed and load per axle of the vehicles. In parallel with this technological evolution, there are also safety issues involving the structural integrity of both vehicles and the railway infrastructure itself, which must be monitored more frequently to guarantee adequate levels of safety [2–4].

When it comes to track and vehicle condition monitoring, two possible approaches regarding sensor placement are possible, that is, installing sensors on the trackside or on

the vehicle side. Track-mounted sensors can provide detailed information regarding track conditions; however, due to the lengthy extension of railway lines, it is usually only feasible to instrument small track sections, limiting this application only to very critical parts of the track. On the other hand, this approach is particularly interesting when it comes to vehicle monitoring since a small instrumented track section may be used to monitor all the passing vehicles [2,5].

Another option to consider is the installation of onboard monitoring systems on railway vehicles. While instrumenting the vehicles directly may not be as cost-effective as installing systems on the track, it offers greater accuracy in detecting vehicle damage. Furthermore, besides being expensive, having sensors on every vehicle is not as economically prohibitive as instrumenting an entire railway track. On the other hand, this approach has a significant economic advantage when it comes to monitoring railway infrastructure. A single instrumented vehicle can provide information on a large track extension, making it a cost-effective solution for monitoring large railway stretches [6,7]. This particular approach is usually known as drive-by, vehicle scanning, or indirect monitoring and was first introduced in 2004 by Yang et al. [8] for assessment of a bridge's natural frequency from the vehicle's response only.

In recent years, many researchers have concentrated their efforts on developing methodologies for detecting damage to transport infrastructure based on data collected by instrumented vehicles. Concerning railway applications, significant advancements were achieved in detecting damage both on the track and in vehicles while, unlike road applications, only a few works involving drive-by methodologies for condition assessment of railway bridges have been developed so far.

This article focuses on presenting the current state-of-the-art on drive-by methodologies applied to railway subsystems. Given the extent of the subject matter constituted by the set of these applications, the authors have decided to divide the discussion into two articles. As mentioned in Part I [9] of this work, which is related to drive-by railway bridges' condition assessment, this second part covers works related to track and vehicle condition evaluation through indirect monitoring methodologies via instrumented trains. Altogether, the two papers comprise applications of drive-by methodologies applied to the main railway infrastructure subsystems, contributing to the existing field of knowledge.

Table 1 summarizes the available literature review articles highlighting the subjects discussed in comparison with those covered by this work. A discussion of the content of these papers is presented in Part I [9]. As can be seen, there is a clear gap in the related literature when it comes to works dedicated to the application of drive-by methodologies to railway subsystems. This is especially true when it comes to bridges, where most of the reviews are focused on highway applications, and only a few studies related to the railway are cited. To the best of the authors' knowledge, no review has been conducted to cover the application of drive-by techniques to all of the three railway subsystems: vehicle, track, and bridge. Therefore, this work aims to fill this gap by presenting a critical state-of-the-art review of drive-by methodologies applied to the three aforementioned railway subsystems.

**Table 1.** Main topics addressed by the state-of-the-art review articles found.

| Reference | | Bridge Subsystem | | Railway Track Subsystem | Railway Vehicles Subsystem |
|---|---|---|---|---|---|
| | | Highway | Railway | | |
| Ward et al. [10] | | -- | -- | ✓ | ✓ |
| Weston et al. [11] | | -- | -- | ✓ | -- |
| Zhu & Law [12] | | ✓ | -- | -- | -- |
| Malekjafarian et al. [13] | | ✓ | -- | -- | -- |
| Li et al. [14] | | -- | -- | ✓ | ✓ |
| Yang & Yang [15] | | ✓ | -- | -- | -- |
| Bernal et al. [16] | | -- | -- | -- | ✓ |
| Yang et al. [17] | | ✓ | -- | ✓ | -- |
| Wang et al. [18] | | ✓ | -- | ✓ | -- |
| Malekjafarian et al. [19] | | ✓ | -- | -- | -- |
| **Present work** | **Part I** | -- | ✓ | -- | -- |
| | **Part II** | -- | -- | ✓ | ✓ |

In this Part II, relevant works applying to railway tracks and vehicles are discussed, highlighting the advantages and drawbacks of each technique, as well as the challenges to be overcome to enable practical applications, especially those regarding environmental and operational interference. Then, an overview of the works presented is carried out, highlighting the current advances achieved by the authors and the most promising future research trends. Finally, with this review work and the indication of new research trends, the authors intend to provide a significant contribution toward enabling practical applications of drive-by methodologies to rail transport systems.

## 2. Overview of Drive-by Condition Assessment of Railway Track and Vehicle

The dynamic responses of both the vehicle and the track are closely linked and rely on the physical and geometric properties of each subsystem's components. In terms of track components, factors such as the mass, stiffness, and damping of its main components and their interfaces, as well as the track's irregularity profile, have a direct impact on the dynamic responses of the vehicles [20]. Regarding the influence of vehicle components, the dynamic responses of the vehicle are directly related to the stiffness and damping of the suspension systems, as well as the mechanical properties of the car body, bogies, and wheelsets [21,22]. Thus, it is expected that damage to the track or vehicle might cause detectable changes in vehicle responses.

The main idea behind assessing the condition of railway vehicles and tracks based on embedded monitoring systems lies in the development of methodologies capable of distinguishing responses associated with a scenario of damage from those associated with a health condition. Adopting this approach, particularly for the track, holds great significance as it enables the monitoring of extensive stretches of track, with just a handful of sensors installed on a limited number of vehicles. Besides the benefits mentioned above, implementing the drive-by concept in practice requires the development of robust signal-processing methods that can accurately differentiate between changes in dynamic responses caused by damage and those induced by external factors, such as environmental and operational disturbances. The acceleration responses measured on the vehicle can be influenced by various factors, such as temperature variations, wind, track irregularities, vehicle speed, and weight, among others. Therefore, it is crucial to have robust damage indicators and distinguish these external interferences from those caused by damage for any detection methodology to be feasible [19,23,24].

Concerning track and vehicle damage detection through onboard instrumentation, several authors have worked on numerical and experimental studies, and many different

methodologies have been proposed. In these studies, the feasibility of detecting different types of damage was assessed as well as different strategies for overcoming the aforementioned operational and environmental interferences. This part of the review article provides a detailed overview of several of these recent studies identified by the authors. The article layout is illustrated in Figure 1 through a comprehensive flowchart. To enhance clarity, the article is divided into subsections, based on the type and/or region of the defect that needs to be located. Specifically, for the identification of track damage, the works were categorized into three subsections (i) track irregularities' assessment, (ii) rail condition-based assessment, and (iii) rail supporting components' condition-based assessment. Similarly, for vehicle damage identification, the article is divided into two subsections: (i) suspension damage identification, and (ii) wheelset damage identification. This clear and concise categorization will provide a structured approach to better understand and analyze the contents presented in the following sections of the article.

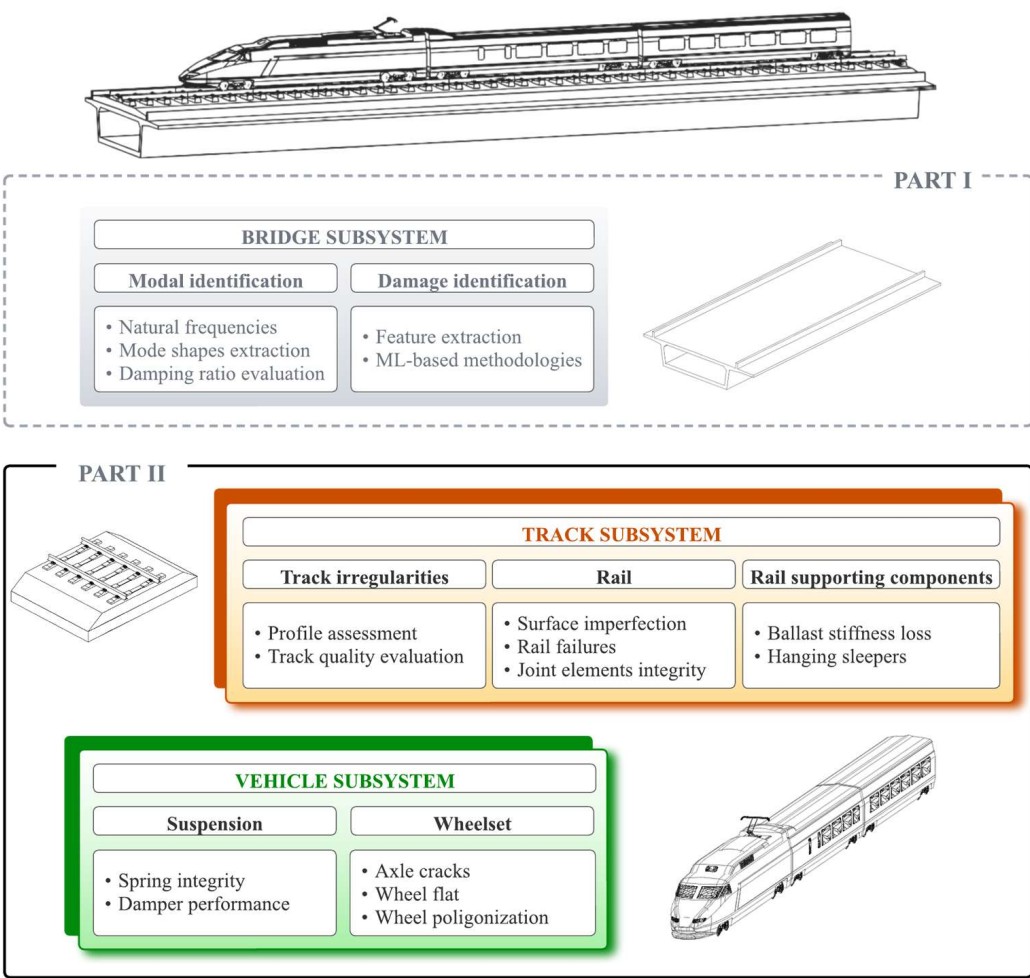

**Figure 1.** Application of drive-by methodologies in railway infrastructures: an overview of the articles' layout.

## 3. Application of Drive-by Methodology on the Railway Track Subsystem

In addition to bridges, monitoring the conditions of the track itself is crucial for rail transport safety. However, given the large extension of railways, this type of inspection is usually performed visually or by dedicated inspection vehicles equipped with several sensors like accelerometers, ultrasonic sensors, computer vision, and laser-based measuring systems [25,26]. However, the high cost associated with the operation of these vehicles

makes it prohibitive for these inspections to be carried out very frequently. Faced with this reality, there is an increasing interest in the development of drive-by methodologies capable of identifying typical defects associated with the track, as well as the irregularity profile of the rails. These methodologies make it possible to take advantage of the high frequency with which regular trains run and ensure earlier detection of damage. In this Section, several works dedicated exclusively to drive-by methodologies for track condition assessment are reviewed.

### 3.1. Track Irregularities Assessment

Track irregularities are characterized by deviations from the ideal rail profile (Figure 2). Typically, these are evaluated in terms of longitudinal level (vertical), lateral alignment, gauge, cross-level, and twist and are strongly correlated with vibrations on the vehicles, having, therefore, a significant influence on their running safety [27,28]. Due to this fact, the level of irregularities of a railway track is an indication of its quality. Usually, these are evaluated by specialized inspection vehicles; however, there has been a growing interest in low-cost alternatives through drive-by methodologies.

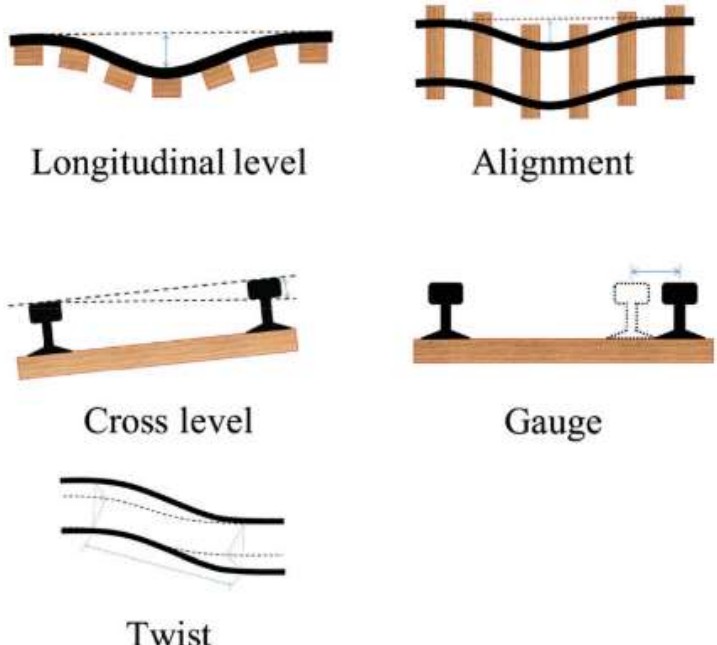

**Figure 2.** Types of track irregularities [29].

In one of the pioneering works in this field, Weston et al. [30] investigated the vertical track irregularity assessment through a set of gyroscopic sensors embedded in railway vehicles. The authors derived analytical expressions correlating the pitch rate of the bogie and axle box vertical acceleration to the track curvature. Grounded on these formulations, a methodology for the vertical irregularity estimation based on a bogie-mounted gyroscope was formulated. The proposed system was installed in an operational train and the results obtained were compared with those derived from acceleration measurements. Good results were achieved for wavelengths greater than 8 m for different vehicle speeds when compared to reference values derived from accelerometers. The irregularities with wavelengths shorter than 8 m could not be accurately estimated by the bogie gyroscopes due to the geometric and dynamic filtering of the bogie and suspensions. Despite leading to less accurate results when compared to accelerometers, the use of the gyroscope has the advantage of being more simple, robust, and capable of providing information over a larger speed range, especially for low speeds, where the low signal-to-noise ratio makes the use of acceleration signals unfeasible. A similar approach, but involving lateral irregularities, was proposed by Weston et al. [31] using a gyroscope to assess the bogie yaw rate and derive

lateral track irregularities. Just as for the case involving vertical dynamics, the formulation for lateral irregularities proved to be accurate and work better for a large speed range compared to the use of accelerometers.

Taking a different approach, which relies on more complex modeling of the dynamic behavior of the vehicle and its interaction with the track, Real et al. [32] assessed vertical track irregularities based on the Frequency Response Function (FRF) between track irregularities and vehicle dynamic responses. This FRF was obtained by taking the Fourier transform of the time domain dynamic equilibrium equations of the train–track interaction system. Then, from the frequency spectra of vehicle acceleration and the application of the inverse FRF, the irregularities in the frequency domain were obtained and converted to the spatial domain by the inverse Fourier transform and vehicle speed. The method was experimentally validated and, in general, reasonably good results were obtained. However, some difficulties were encountered in accurately representing the peaks, mostly due to measurement errors, such as low-frequency components and phase shifts, as well as variations in vehicle speed. Instead of using a frequency domain formulation, Odashima et al. [33] proposed the use of a state space model in conjunction with a Kalman filter to estimate track vertical irregularities and a 10 m-chord versine (perpendicular distance from the chord to the curve). The method was numerically and experimentally validated and reasonably good overall results were obtained for the track geometry, except for the larger wavelength components. On the other hand, the 10 m-chord versine was estimated with very good accuracy both numerically and experimentally. A similar approach using Kalman filters was also applied by Dertimanis et al. [25] in which very precise results were obtained numerically; however, there was no experimental validation.

A comparison between time domain and frequency domain-based formulations for assessing alignment and cross-level irregularities was conducted by De Rosa et al. [34]. Two formulations based on state space models (time domain) were investigated, one grounded in the use of Kalman filters and the other on the Unknown Input Observer (UIO) technique. Additionally, one frequency domain approach based on the inversion of FRFs was investigated. The numerical studies performed demonstrated that all methods have comparable performance, except for the estimation of cross-level, where the UIO had worse performance. Additionally, it was demonstrated that both the frequency domain and the Kalman filter-based technique performed very well under high levels of measurement noise.

Still concerning model-based methods, Obrien et al. [35] proposed the use of the cross-entropy optimization algorithm to determine the vertical irregularities profile, which minimizes an objective function composed of the squared sum of the differences between measured and simulated accelerations. The method was tested for a very simplified 2 DOF railway vehicle quarter-car model running over an infinitely rigid track, for which very accurate results were obtained. The formulation was later enhanced by Obrien et al. [36] considering a 10 DOF 2D vehicle model including the car body, two bogies, and four wheelsets, as well as a three-layer flexible track model. It has been numerically demonstrated that, in the absence of sensor noise, the irregularity profile is estimated with great accuracy. However, the presence of noise is responsible for generating drifts on the estimated profiles and convergence issues on better-quality tracks. Furthermore, the presented methods, which are based on numerical models to estimate irregularities, are very dependent on their quality. Obtaining a good quality model requires precise knowledge of the physical properties of the track and the vehicles, which is always a challenge when it comes to practical applications. A field validation of the method presented by OBrien et al. [36] was conducted by the same authors [37] during a one-month experimental campaign performed on an Irish Rail intercity train operating regularly between Dublin and Belfast. A triaxial accelerometer and a gyro-meter were installed in the bogie of the wagon (Figure 3), seeking the position as close as possible to the center of mass. From data collected by these sensors, the longitudinal level of a 200 m stretch of track was inferred by the methodology based on the cross-entropy optimization algorithm [36] and results were compared with those derived from conventional measurement techniques. Generally, it was possible to

verify a good agreement between the shape of the inferred profiles and the reference values; however, the magnitude was severely overestimated. The main reasons pointed out for this error were: (i) errors in the gyro-meter calibration; (ii) the use of a 2D model, as well as calibration inaccuracies of this model, and (iii) the fact that the track was not loaded during the measurement of the profile used as reference.

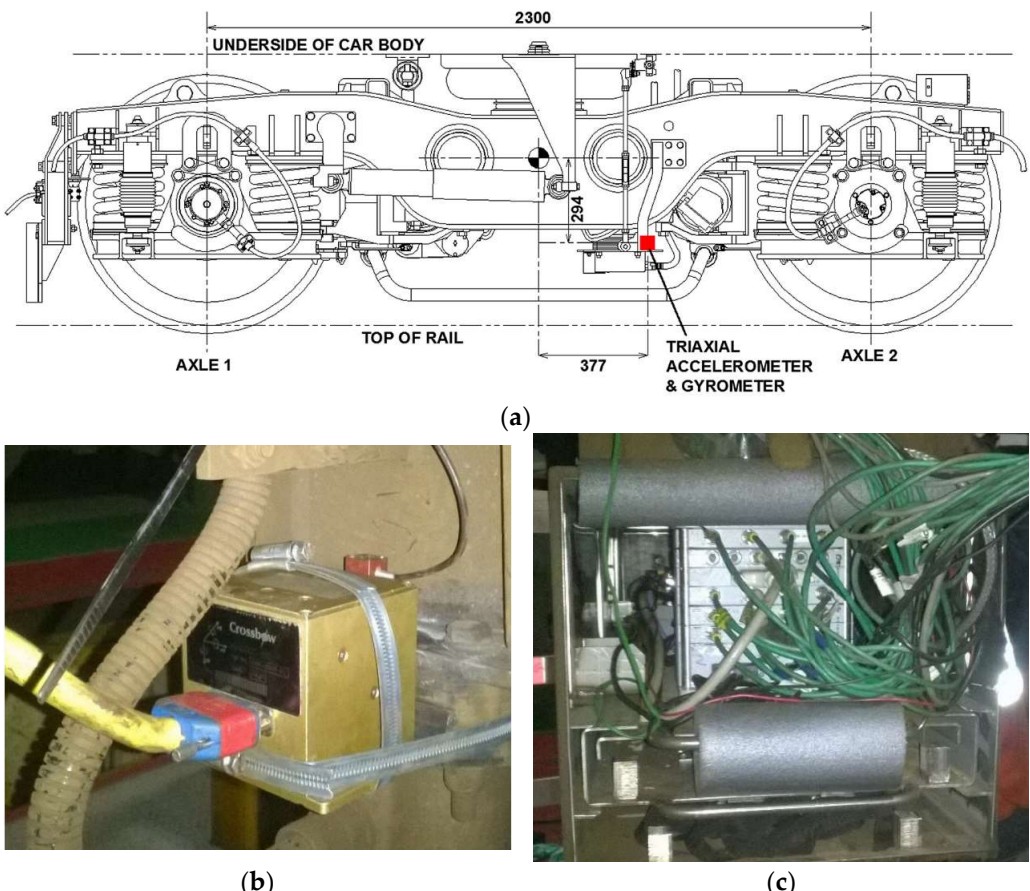

**Figure 3.** Experimental setup for longitudinal level assessment: (**a**) positioning; (**b**) accelerometer and gyro-meter installation and (**c**) data acquisition system [37].

More recently, with the increasing use of deep learning techniques, some authors have chosen to train this type of model to infer track irregularities based on vehicle accelerations. Li et al. [38] trained an extended auto-encoder (EAE) model to predict track longitudinal level from car body vertical acceleration using both simulated and real data. The architecture of the model is presented in Figure 4a. During the training phase, the encoder is used to compress the irregularity information and create a latent representation that can be later reconstructed by the decoder. The estimator, in this case, a neural network, uses car body acceleration to estimate this latent representation learned by the encoder. As demonstrated in Figure 4b,c the model can accurately estimate the longitudinal level for most wavelengths even for real-world data. Wavelengths shorter than 5 m, however, could not be captured due to the filtering effect of the suspensions and were removed with a high pass filter. Hao et al. [39] also made use of deep learning techniques for assessment of the longitudinal level of high-speed railway lines in China. The authors proposed a NN model combining an Attention Mechanism (AM), a Convolution Neural Network (CNN) and a Gated Recurrent Unit (GRU), which was trained using measured data from a specialized inspection vehicle. The inputs for the model are the vertical and lateral acceleration measured at the car body and the vehicle speed, and the output is the track longitudinal alignment. Good results were obtained when compared to measured irregularities; however, just as in the previous study, the filtering effect of the suspensions makes it impossible to assess low

wavelength irregularities. In this case, the author succeeded in estimating wavelengths above 3 m.

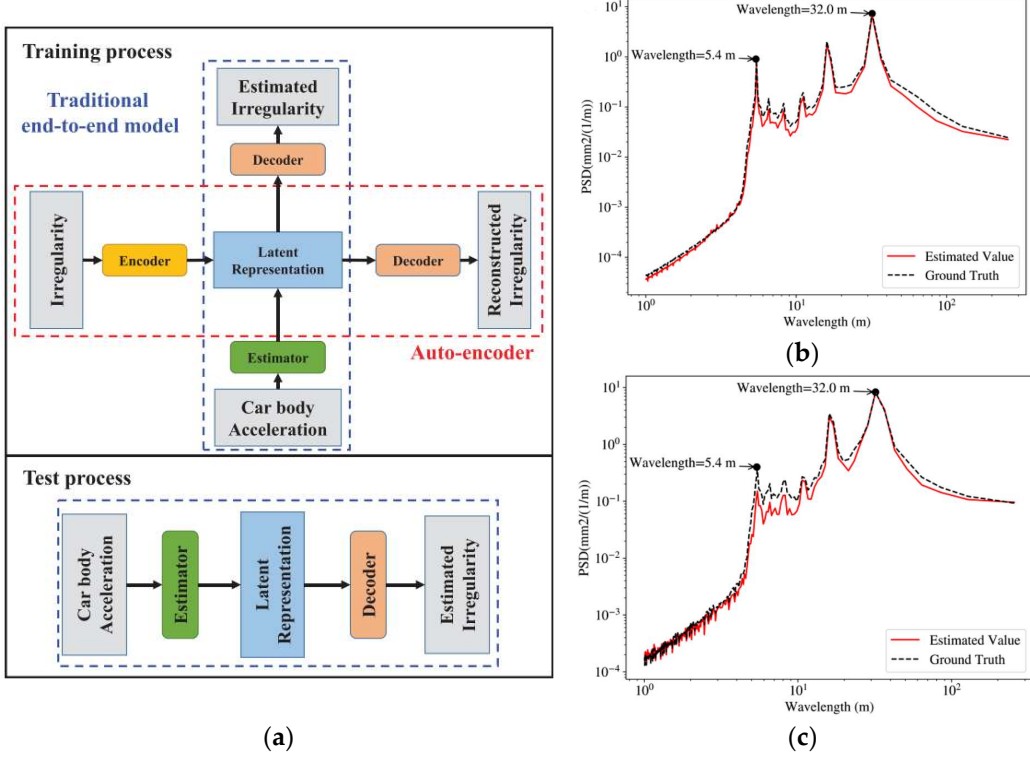

**Figure 4.** Track longitudinal level prediction based on EAE: (**a**) EAE architecture, (**b**) results for the simulated training set, and (**c**) results for the real training set [38].

Unlike previous works that focused on obtaining the profile of irregularities, La Paglia et al. [40] took advantage of the close relation between vertical irregularities and bogie vertical acceleration, in order to train a linear regression model capable of inferring track quality from the RMS value of bogie acceleration. Two models were trained, one to estimate the RMS value of vertical alignment, and the other to obtain its peak value, where greater accuracy was obtained for windows of 100 m and 25 m, respectively. As expected, the estimation of the peak results was not as accurate as that of the RMS values, although it presented reasonable precision to infer the quality of the track. Tsunashima [29] also proposed a method based on RMS values but using the combined information of vertical and lateral car body acceleration and car body roll rate. Based on these data a machine learning algorithm, particularly the support vector machine (SVM), was used to detect track faults. The methodology was validated both numerically and experimentally on an operational train, and the degradation associated with the longitudinal level, alignment, and cross irregularities was successfully identified. Tsunashima & Hirose [27] used the maximum RMS value of car body vertical acceleration, calculated for small track sections, as an indicator of track fault. For track segments identified as critical, a detailed time-frequency analysis, based on the Hilbert Huang Transform, was used to classify the type of defect. The longitudinal level degradation was shown to be associated with frequencies below 5 Hz, making it distinguishable from localized defects, which are more related to a higher frequency range. In an investigation involving similar track faults, Tsunashima & Takikawa [41] demonstrated the feasibility of applying Convolutional Neural Networks directly to the time-frequency color map representation to automatically identify and distinguish between these defects.

Still concerning the use of machine learning algorithms for general track quality assessment, De Rosa et al. [42] investigated the performance of three different classifiers (the decision tree and SVM associated with linear and Gaussian kernel functions) on

inferring the standard deviation of alignment and cross-level track irregularities. The models were trained based on a simulated data set where the standard deviation of lateral and roll bogie frame accelerations were used as inputs and the track class is given as output. After training, the algorithms were tested based on measured data, and the SVM, in both versions, presented a better overall performance for this particular application compared to the decision tree. On the other hand, the decision tree was more conservative compared to the other algorithms.

Some issues regarding practical applications of track condition assessment were addressed by Malekjafarian et al. [43] during an experimental study in which the amplitude of the analytical signal, derived from the Hilbert transform of the bogie acceleration response, was used for track damage assessment. The speed of the train, which could be measured but not controlled, was decisive for the effectiveness of the method, and its influence was evaluated in detail. The authors found that, at speeds below 50 km/h, the energy levels were insufficient to provide any conclusive results. In addition, the significant differences in energy for signals collected at different speeds proved to be a difficult factor in the identification of damage. To correct this issue, the authors proposed the use of a normalization factor linearly dependent on the vehicle speed. Data from several train passages were evaluated and the regions with high energy in the signals corresponded quite well with zones identified as degraded by a specialized inspection vehicle.

### 3.2. Rail Condition-Based Assessment

Acceleration measurements on vehicles can provide good indications of the presence of localized defects that affect the surface geometry of the rails, such as Squats and Crushed Heads, which are common defects associated with rolling contact fatigue (RCF). A comprehensive review of these RCF faults is presented by Magel [44]. This type of defect causes impact excitations in the wheel–rail contact, which can be identified by the acceleration measurements. Li et al. [45] conducted studies aimed at determining the causes and evolution of squat-type defects. The dimensions of several defects of this type found in the Dutch railway network were analyzed. Based on the typology of these defects, numerical simulations of the passage of the wheel over the defects were carried out, through which it was demonstrated that there is a close relationship between wavelengths in the contact forces and the dimensions of the squats. Based on these studies, it was predicted that Squat initiation and growth are typically associated with certain wavelengths between 20 and 40 mm. These predictions were further validated experimentally by Li et al. [46], who used the Wavelet transform of acceleration records, measured at the axle box of a railway vehicle traveling at 110 km/h, to identify Squat related frequencies. The use of the Wavelet transform instead of the Short Time Fourier Transform (STFT) provides better time and frequency resolutions. The robustness of this signal processing technique for track damage identification was investigated in depth in the pioneering work developed by Caprioli [47], and its advantages compared to STFT.

In Figure 5, a Squat defect and the associated axle box Wavelet acceleration spectrogram are presented. As can be seen, the presence of the defect results in an increased level of energy around high frequencies, especially around 1500 Hz, which is consistent with the wavelengths found to be associated with Squats in the work conducted by Li et al. [46].

Based on the aforementioned investigations [45,46], Molodova et al. [48] proposed an automatic methodology for Squat detection based on three-dimensional acceleration measurements taken at the four-axle boxes of a railway bogie (Figure 6). To enable practical applications, the authors used signal filters for removing noise from the data and proposed a procedure for eliminating the contribution of wheel defects based on revolution period, which is a function of the wheel radius and wagon speed. Then, automatic damage detection is performed based on an empirically defined threshold value for the Wavelet power magnitude in the frequency range associated with squats. With the proposed methodology, the authors achieved a 100% hit rate for moderate and severe squats and 78% for small defects. The method, however, is sensitive to track discontinuities, such as welds

and joints and vehicle speed. Regarding vehicle speed, and different from the applications related to bridges, higher speeds seem to be beneficial for the identification of rail faults, as they result in greater vibration energy when the wheel passes over a defect. The effect of the vehicle speed was investigated in greater detail in parametric studies conducted by Molodova et al. [49], where a relation between speed and axle box acceleration at squats was derived. Additionally, the authors demonstrated that the frequencies at squats are somehow linked to the natural frequency of the track. This particular study raised important aspects to be addressed for the practical application of squat detection. Given the dependence of the responses associated with Squats to the speed and natural frequency of the rail, it is essential to establish a precise mapping between these quantities to support automatic identification algorithms.

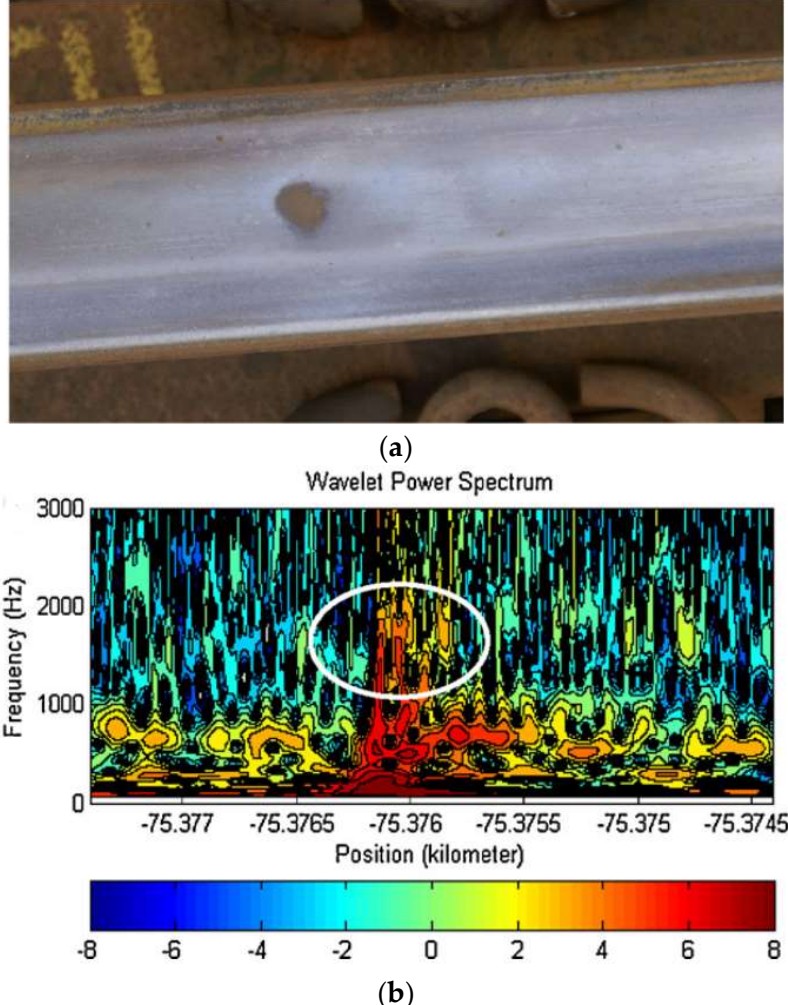

**Figure 5.** Squat type defect identification: (**a**) Top view of a damaged rail and (**b**) Wavelet acceleration spectrogram [46].

The application of the Wavelet transform as a feature for superficial track damage detection was also explored by Cantero & Basu [50]. Based on a simplified 2 DOF multibody vehicle model, the authors conducted simulations of the vehicle passing at 180 km/h over a bump-type track defect. As can be seen in Figure 7, the passage of the wheel over the bump, which happens at approximately 30 s, results in energetic components which are spread along frequencies higher than those associated with the healthy track. Based on these findings, an automatic damage detection algorithm was proposed by defining a threshold for the value of the sum of all Wavelet coefficients for each time instant. This powerful time-frequency analysis technique was also used by Oregui et al. [51] to evaluate

the bolt tightness of rail joints. The correlation between the energy content calculated from the Wavelet transform of axle box acceleration for specific frequency ranges and bolt tightness of rail joints was established employing a series of controlled experimental tests comprising different bolt tightness. Based on this, a simple algorithm was proposed that could efficiently distinguish between tight, intermediate loose, and completely loose bolts. The method was experimentally validated by axle box acceleration measured on a regular service tram with resilient wheels, which poses more challenges for these methods, given the lightness of the vehicle, low speed, and flexibility of the resilient wheels. Even so, the algorithm performed quite well in detecting the state of the rail joints.

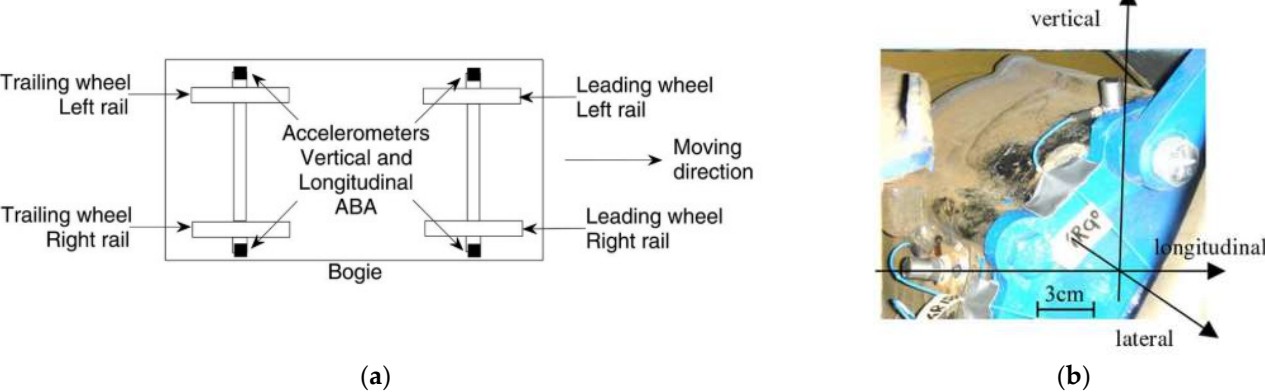

**(a)**　　　　　　　　　　　　　　　　　　　　　　　**(b)**

**Figure 6.** Measurement setup for Squat detection: (**a**) Sensors positioning in the bogie and (**b**) detail of the accelerometer's installation at the axle box [48].

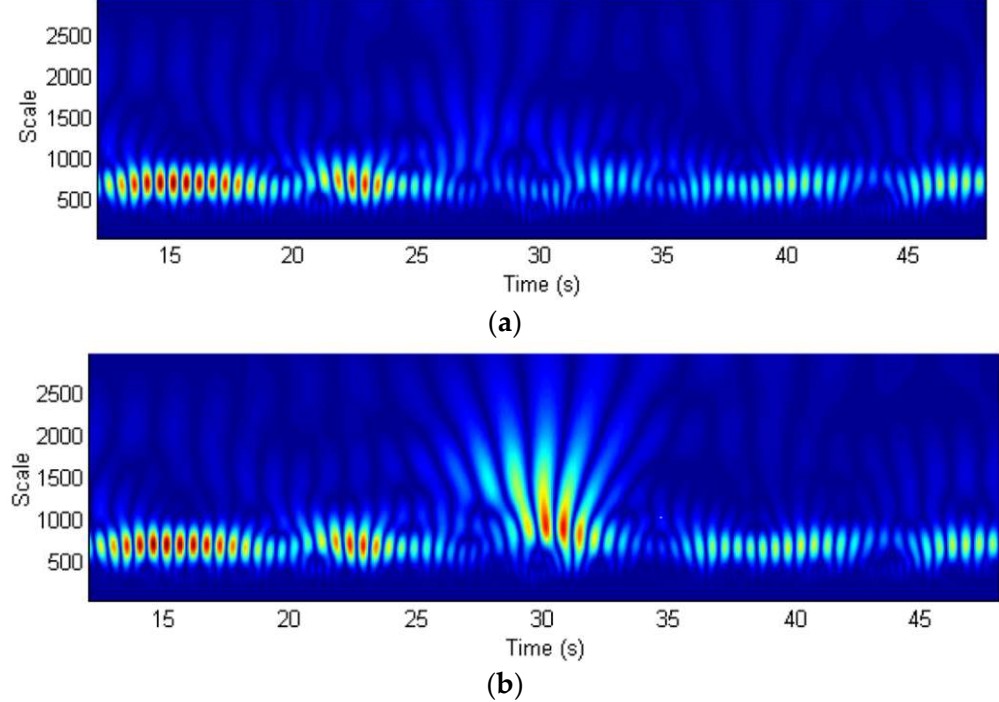

**Figure 7.** Wavelet transform of vehicle acceleration: (**a**) healthy track and (**b**) damaged track [50].

Another undesirable condition affecting the surface geometry of the rails is corrugation. This undulatory defect consists of short-length waves distributed along the top of the rail that may cause strong forces at the wheel–rail contact, as well as high noise levels [52]. Xie et al. [53] proposed a machine-learning (ML) methodology to identify and characterize rail corrugation in metro lines. Firstly, a 1DCNN was trained to identify, as well as classify, rail corrugation by its wavelength from acceleration measured at the axle

boxes. Then, based on numerical simulations at different speeds and rail corrugation of different wavelengths and depths, a Kriging surrogate model (KSM) was constructed to model the relationship between vehicle speed, corrugation depth, and axle box RMS acceleration levels. From this model, the corrugation depth could then be inferred by reverse optimization through a Particle Swarm Optimization (PSO) algorithm. The method was experimentally validated and an accuracy of 99% was achieved for the fault identification and an average relative error of 6.75% for the corrugation depth estimation. Xiao et al. [54] investigated the use of a different ML algorithm to identify rail corrugation on heavy-haul railway lines from axle box acceleration. The acceleration signal was decomposed by the wavelet packet transform (WPT) and an adaptive short-time Fourier transform was used to compute the frequency content of the packets with the best resolution. Then a damage feature based on the mean Renyi entropy value of the packets is proposed and the classification is performed by a Support Vector Machine (SVM) ML algorithm. Experimental validation was performed on a real train and an accuracy of 93% was achieved.

Taking advantage of the high acoustic emissions associated with corrugations, some authors have investigated the possibility of using embedded microphones instead of accelerometers to detect corrugations. Liu et al. [55] used sound pressure levels measured by a bogie-mounted microphone (Figure 8) to detect corrugation on metro lines. Based on experimentally validated models, the authors performed simulations of the wheel-rail noise for different corrugation levels and applied the Wavelet Packet Transform (WPT) to decompose the acoustic signal. The energy levels of relevant parts of the decomposed signal regarding corrugation detection were used to assess the corrugation amplitude. The simulated operational conditions were then used for fitting a KSM model establishing a relationship between acoustic energy, corrugation amplitude, and corrugation wavelength. The model was validated based on field tests, and the predicted amplitudes were compared with measured ones, resulting in an amplitude relative error of 10.5%. Wei et al. [56] demonstrated that it is also feasible to detect corrugations from noise measured inside the car body of a high-speed train. Based on experimental investigations, it was concluded that corrugations are usually correlated with noise frequencies between 400 and 700 Hz. A corrugation indicator was then proposed based on the scale-averaged wavelet power, computed for the aforementioned frequency range, and the wavelength was derived from the PSD of the noise measured at the track sections with corrugation. A comparison with traditional methods showed that the proposed methodology can accurately identify corrugation. The possibility of installing the microphone inside the car body is very attractive, since the sensor can be very well protected; however, its performance in more general conditions, such as a train full of passengers talking, still needs to be demonstrated.

One aspect hardly addressed in most works involving damage to rails is the comparison of the performance of different damage features against various interferences. Lederman et al. [57] conducted parametric studies comparing the influence of position error from the GPS sensor, vehicle damping, and vehicle natural frequency on the identification accuracy achieved, using a data-driven methodology and four different damage features: (i) temporal-frequency, (ii) spatial-frequency (iii) spatial amplitude, and (iv) signal-energy (spatial squared amplitude). Some of the results of these studies are presented in Figure 9 for the identification of a broken rail. As can be seen, the signal energy was the most accurate feature among all the tested conditions. In addition, from this figure it is also possible to conclude that the position error is the parameter that most impacts the results and the signal-energy (spatial squared amplitude) feature is quite immune to this effect. This finding reaffirms the need to have reliable position systems for the success of data-driven approaches. Yang et al. [58] demonstrated that the squared amplitude computed from the signal's envelope derived from the Hilbert transform is also a very powerful feature for rail damage identification.

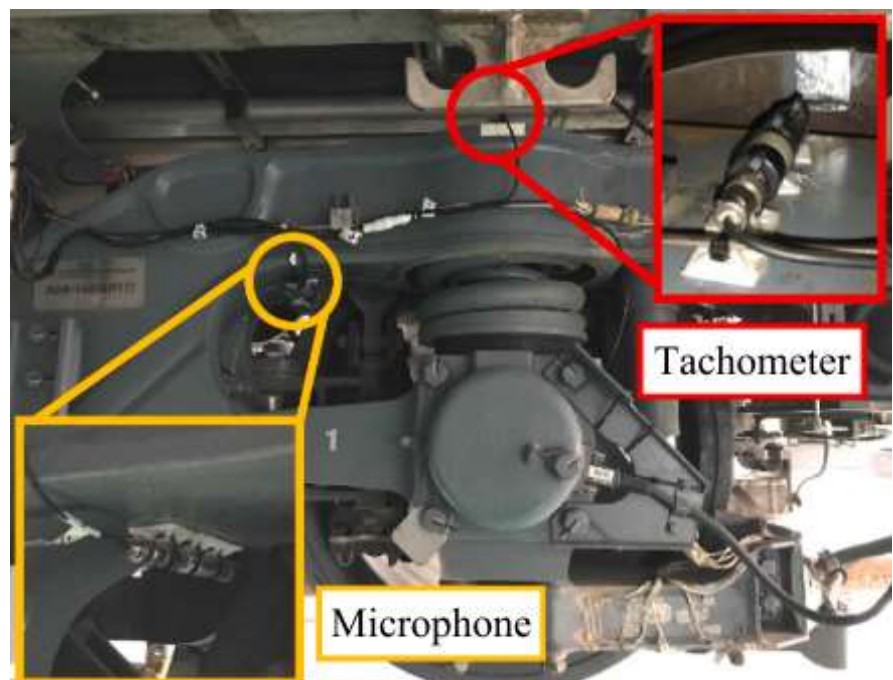

**Figure 8.** Bogie sound pressure level measuring system [55].

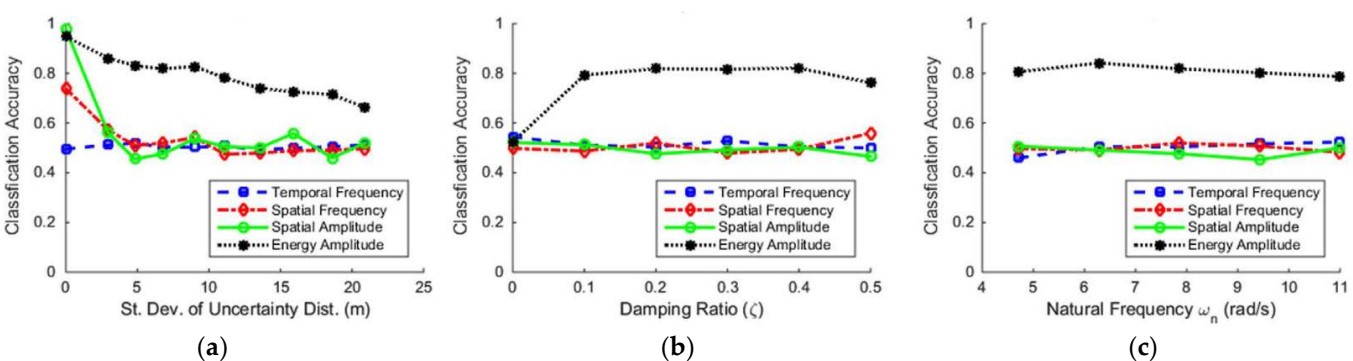

**Figure 9.** Classification accuracy for a broken rail as a function of: (**a**) position error standard deviation, (**b**) vehicle damping ratio, and (**c**) vehicle natural frequency [57].

### 3.3. Rail Supporting Components' Condition-Based Assessment

Differently from previous works, which are related more to rail defects, Quirke et al. [59] developed a drive-by methodology for detecting damages to the track's interlayers through the assessment of variations in the global track stiffness. The authors used the Cross-entropy optimization algorithm to infer the stiffness profile, which minimizes the differences between a measured and a simulated bogie acceleration response. The methodology was tested for a simplified 2D half-bogie vehicle model and a 2D single-layer model. Good results were obtained for reasonably high speeds (120 km/h) under various levels of measurement noise and uncertainties in vehicle properties.

Zhang et al. [60] studied the behavior of the natural frequencies of the train track interaction system and analytically demonstrated that, as the vehicle moves along the track, the natural frequencies of the system vary in a well-defined periodic way. However, in the case of a damaged rail-supporting element, this regular behavior is changed and a noticeable shift in the natural frequencies is verified. Based on these findings, a damage index calculated from this frequency shift was proposed for detecting stiffness reductions on track supporting elements. The proposed index was validated through numerical simulations employing very simplified 2D models and reduced-scale experiments. During

these validations, promising results were obtained; however, the performance of the method in dealing with more complex situations, inherent to practical applications, still needs to be checked.

Yang et al. [61] investigated the use of the Instantaneous Amplitude Squared (IAS), derived from the Hilbert transform of the wheel–rail contact response, as a feature for identifying damage at the track foundation. Through a sensitivity analysis, the authors concluded that the driving frequency, which is the frequency component directly proportional to the vehicle speed, has an amplitude greatly influenced by foundation damage. Based on this conclusion, the contact response was calculated from the driving frequency component by a back calculation procedure based on vehicle dynamic responses. The performance of the IAS was evaluated for the simplified vehicle and track models, presented in Figure 10, in which all the components of the foundation were modeled by one spring layer. The influence of several field factors was investigated and it was concluded that the accuracy of the method is reduced by high levels of irregularities and also by both very high and very low speeds, with the authors obtaining better performance at 150 km/h. The use of the IAS as a damage feature was also explored by Xiang et al. [62], who applied the method not only for foundation but also for rail damage detection. The authors also investigated the influence of other factors besides those evaluated by Yang et al. [61] and were able to conclude that the damping of the vehicle suspension is unfavorable to the accuracy of the method, while the mass is beneficial. This same feature was also used by Yang et al. [58] in a more challenging situation involving a track section over a viaduct. Even for this case, the IAS was shown to be sensitive to damage at the tracks interlayer.

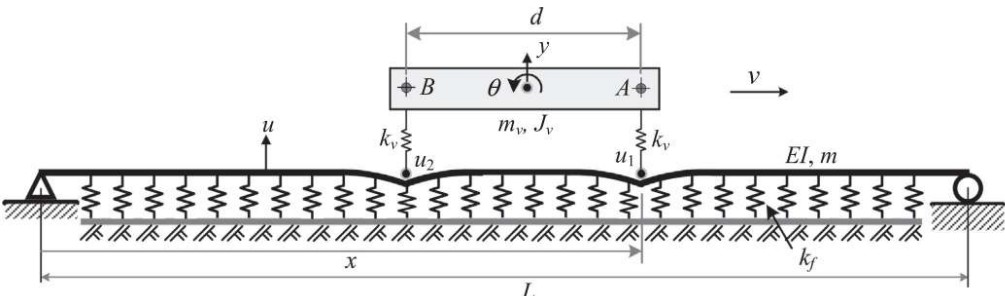

**Figure 10.** 2D interaction model for track foundation damage assessment [61].

With a different approach, but still based on contact point response, Yang et el. [63] proposed the use of variations in the first rail frequency as a feature for foundation damage. To avoid undesirable influence from vehicle-related frequencies, the FFT of the contact response was used for the rail frequency assessment. To include spatial information for the location of the damage, it was proposed to divide the track into 20 m stretches, for which the spectrum was calculated individually. The feasibility of the method was demonstrated based on a simplified vehicle model composed of a sprung mass and the track modeled as a beam on an elastic foundation. Promising results were obtained but validation against more field factors and real field tests is still needed.

Still concerning damage to track interlayers, a recent study by Malekjafarian et al. [64] further explored the detection of damage associated with track loss of stiffness, particularly hanging sleepers, through the use of the filtered bogie displacement (FBD) response calculated by the integration of acceleration signals. Firstly, a numerical study, based on a 2D 10 DOF vehicle model and a three-layer track model, was conducted to demonstrate the good correlation between the FDB and the track's loaded profile. These numerical findings were then experimentally confirmed by measurements taken on a passenger train. Then, the authors demonstrated that a localized track loss of stiffness results in a bump on the displacement signal, which can be used as a damage indicator. The methodology was also tested considering uncertainties in the train speed and measurement noise, and despite the amplitude of the FBD being significantly affected by these parameters, it was demonstrated

that it can still be used as a reliable damage feature as long as the average value of several train passes is considered.

### 3.4. Summary of Discussed Literature on Drive-by Methodologies Applied to the Track Subsystem

In Table 2, a summary of the main references discussed in the previous sections regarding the application of drive-by methodology on the railway track subsystem is presented. Aspects such as the type of damage detected, study type, and main techniques used are highlighted for a better consolidation of the contents presented.

**Table 2.** Summary of the presented works in drive-by methodology on the track subsystem.

| References | Scope | Study Type | Methodology |
|---|---|---|---|
| Weston et al. [30,31] | TI | A/E | Analytical expressions between dynamic responses and irregularities for extracting the profile of irregularities based on measurements of gyroscopes and accelerometers in the bogies. |
| Real et al. [32] | TI | A/E | Inverse modeling based on frequency response functions between dynamic responses and irregularities to extract irregularity profiles. |
| Odashima et al. [33] | TI | N/E | Inverse modeling based on Kalman filters and state space models to extract irregularities |
| Dertimanis et al. [25] | TI | N | Inverse modeling based on Kalman filters and state space models to extract irregularities |
| De Rosa et al. [34] | TI | E | Two formulations in the time domain, one based on the use of Kalman filters and the other on the Unknown Input Observer, and one frequency domain approach based on the inversion of FRFs for lateral alignment assessment |
| Obrien et al. [35,36] | TI | N | Reverse modeling based on cross-entropy optimization algorithm to obtain the profile of irregularities that makes the simulated response closer to the actual responses |
| Obrien et al. [37] | TI | E | Reverse modeling based on cross-entropy optimization algorithm to obtain the profile of irregularities that makes the simulated response closer to the actual responses |
| Li et al. [38] | TI | N/E | Using extended auto encoders to find a latent representation of irregularities based on measured accelerations |
| Hao et al. [39] | TI | N/E | Deep neural network model to infer longitudinal alignment profile based on measured responses |
| La Paglia et al. [40] | TI | E | Linear regression model between RMS acceleration on bogies and RMS irregularities |
| Tsunashima [29] | TI | N/E | Use of a support vector machine to classify irregularities severity from RMS acceleration on bogies |
| Tsunashima & Hirose [27] | TI | N/E | Identification of potentially damaged areas by RMS acceleration at the bodies and detailed investigation of these areas by Hilbert Huang's Transform |
| Tsunashima & Takikawa [41] | TI | N/E | Recognition of damage patterns in Wavelet time-frequency representation trough convolutional neural networks |
| De Rosa et al. [42] | TI | N/E | Support vector machines and decision trees to classify faulty conditions from the standard deviation of lateral and rolling accelerations of bogies |
| Malekjafarian et al. [43] | TI | E | Instantaneous amplitude derived from the analytical signal calculated using the Hilbert transform |
| Li et al. [45] | RC | N | Time-frequency representation derived from Wavelet transform of axle box acceleration |
| Li et al. [46] | RC | E | Time-frequency representation derived from Wavelet transform of axle box acceleration |
| Caprioli [47] | RC | E | Performance comparison between time-frequency representation derived from Wavelet transform and short-time Fourier transform STFT of axle box acceleration in rail condition assessment |
| Molodova et al. [48] | RC | E | Use of the Wavelet power magnitude in the frequency range associated with the squat as a damage feature |

**Table 2.** *Cont.*

| References | Scope | Study Type | Methodology |
|---|---|---|---|
| Molodova et al. [49] | RC | N | Parametrical study based on numerical simulations to assess the influence of vehicle speed and track natural frequency on squat-related frequencies at the axle box. |
| Cantero & Basu [50] | RC | N | Use of the magnitude of the sum of wavelet transform coefficients as a damage feature |
| Oregui et al. [51] | RC | E | Power spectrum derived from the Wavelet transform associated with classification algorithm based on several condition statements for joint bolt tightness estimation. |
| Xie et al. [53] | RC | N/E | Use of a 1D convolution neural network based on acceleration measurements on axle boxes for corrugation identification and classification by wavelength and a kriging surrogate model for corrugation depth estimation |
| Xiao et al. [54] | RC | E | Renvi entropy of the packets resulting from a signal decomposition by the wavelet packet transform and a support vector machine for corrugation classification. |
| Liu et al. [55] | RC | N/E | Kriging surrogate model built based on the energy of corrugation-relevant packets after applying wavelet packet transform to sound pressure recordings |
| Wei et al. [56] | RC | E | Signals power derived from the Wavelet spectrum for the frequency range of acoustic emission associated with corrugation |
| Lederman et al. [57] | RC | N/E | Rail condition assessment based on several features, namely (i) Time-frequency representation, (ii) Distance-frequency representation; (iii) Distance-amplitude representation; and (iv) Signal Energy. |
| Yang et al. [58] | RC/RS | A/N | Square of the instantaneous amplitude derived from the Hilbert transform (IAS) |
| Quirke et al. [59] | RS | N | Reverse modeling for derivation of the track's stiffness profile by the application of the cross-entropy optimization algorithm |
| Zhang et al. [60] | RS | A/N | Use of variations in the natural frequency of the train track interaction system |
| Yang et al. [61] | RS | A/N | Square of the instantaneous amplitude derived from the Hilbert transform (IAS) |
| Xiang et al. [62] | RC/RS | A/N | Square of the instantaneous amplitude derived from the Hilbert transform (IAS) |
| Yang et al. [63] | RS | A/N | First natural frequency of rail vibration derived from the Fourier transform of contact point responses |
| Malekjafarian et al. [64] | RS | N/E | Discontinuities in the filtered bogie displacement response calculated by the integration of acceleration signals measured at the bogies |

Abbreviations: Track irregularities (TI), rail condition (RC), rail supporting elements condition (RS), analytical (A), numerical (N), and experimental (E).

## 4. Application of Drive-by Methodology to the Railway Vehicle Subsystem

When it comes to vehicle monitoring, the advantageous scenario provided by the drive-by concept, in which a single vehicle is capable of collecting information along the entire route, is now reversed. Given this situation, in many cases works focus on identifying damage or vehicle properties through sensors on the railway track, which are capable of monitoring a whole set of vehicles [65,66]. This type of approach, of course, is suitable for detecting damages that affect the structure of the vehicle and have an impact on its dynamic behavior, which is the major focus of this work. Even so, one can find some works in the literature involving the use of onboard monitoring systems in the detection of faults in railway vehicles, mainly related to suspension and wheel damage, which affect its dynamic behavior. This section is dedicated to presenting some of these methodologies focused on global vehicle dynamic behavior.

### 4.1. Suspension Damage Identification

Tsunashima et al. [67] proposed the use of a multiple-model-based approach to the assessment of suspension faults on a railway wagon. The vehicle was modeled grounded

on multibody formulations and several different models were proposed, each one considering a different type of suspension failure, and the failure of a particular measurement equipment was also considered. The damage identification was then performed by finding, through a Kalman Filter, the model that best represents the measured outputs over time, which in this particular case were bogie and car body lateral acceleration and the yaw rate of the bogie. The performance of the method was evaluated for two failure types, a faulty side damper, and a malfunctioning lateral bogie accelerometer. The method has proven its potential to be an efficient alternative for detecting suspension damage; however, its performance still needs to be verified under the influence of environmental and operational disturbance inherent to practical applications. A similar approach was adopted by Liu et al. [68], in which the recursive least square (RLS), an adaptive filtering algorithm closely related to the Kalman Filter, was used to estimate the stiffness and damping constants of a railway vehicle from dynamic measurements taken at the bogie and axle boxes. The validations based on simulated and measured data revealed that the RLS is capable of providing reliable estimates under noise and uncertainties in the parameters.

The accuracy of model-based approaches, such as those previously presented, tend to strongly rely on high-quality numerical models, which are not always available and may impair the accuracy of the method. Wei et al. [69] performed a comparative study comparing some common model-based formulations and data-driven ones: the robust observer, the Kalman filter combined with the generalized likelihood ratio test method, the dynamical principal components analysis, and the dynamical canonical variate analysis. All approaches were applied to the suspension fault detection problem, and the authors concluded that, in general, data-driven methods tend to outperform model-based methods. Thus, some authors have tried to overcome this difficulty by proposing data-driven approaches. Mei & Ding [70] demonstrated analytically that under normal operating conditions, that is when all the springs and dampers of a bogie have the same properties, the bounce and pitch movements of the bogie are uncoupled. Based on this, the authors proposed the use of cross-correlation functions to identify possible couplings between these movements that might indicate the existence of damage in some suspension components. Dumitriu et al. [71] investigated the RMS values of bogie acceleration under normal operating conditions and for a primary suspension damper failure. A simulated and measured dataset of the vehicle without any damage was used to establish a zone associated with the normal operating condition, and values outside this zone were assigned to damage scenarios. To test the methodology, several damage scenarios were simulated and the method was capable of detecting damage causing a reduction of 50% or more in the damping coefficient.

A more complex approach to identifying different types of suspension faults, i.e., anti-yaw dampers fault (YDF), lateral dampers fault (LDF), and yaw & lateral dampers fault (Y&LDF), was developed by Ye et al. [72]. The multiscale permutation entropy (MPE) algorithm was used as a feature extractor; this algorithm consists of a group of Permutation Entropy (PE) values that capture randomness and dynamic changes of the time series. Thus, a Manifold learning algorithm, particularly the linear local tangent space alignment (LLTSA), was used to reduce the dimensionality of the feature matrix generated by the MPE. Finally, a Least Squares Support Vector Machine (LSSVM) model was trained to classify the between scenarios of normal operation, YDF, LDF, and Y&LDF. The method was validated numerically and experimentally and was able to precisely classify the damages. Furthermore, a comparison between the performance of the LLTSA and the Principal Component Analysis (PCA) was performed, and the LLTSA exhibited better performance against the non-linearities inherent to railway systems. The steps of the proposed methodology are depicted in Figure 11.

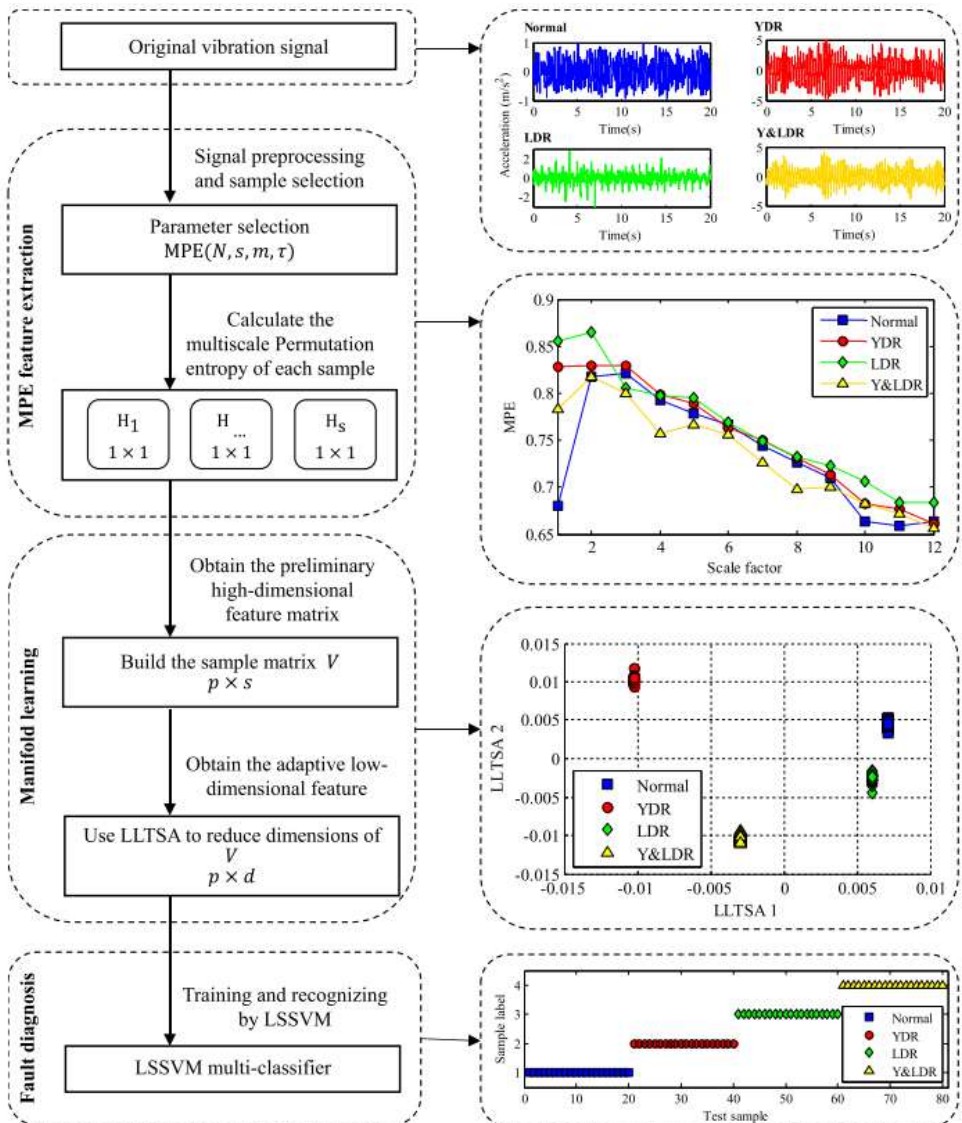

**Figure 11.** Methodology for suspension fault identification [72].

More recently Ye et al. [73] trained a 1 DCNN model to extract features associated with suspension damage and, based on these features, a fully connected NN was used to classify the same suspension faults addressed by the previous study. In this study, the influence of different track irregularities and the state of wear on train wheels on the model accuracy was investigated. To mitigate the irregularities interference, a certain level of white Gaussian noise was added to the signals, to make them all have high-frequency content generated by impacts with track irregularities. Regarding wheel wear, simulations for several wear states were included in the training sets. During validation, these strategies proved to be crucial for the success of the methodology, which was able to classify with great precision the different types of damage for simulated and measured data.

### 4.2. Wheelset Damage Identification

Another field of application for onboard monitoring systems involves the detection of damage related to the wheelset components, more specifically the axle and the wheels. Regarding axle defects, Hassan et al. [74] conducted a numerical study to verify the feasibility of detecting cracks on a railway axle from displacement signals on several points along the axle. Two different axles, with hollow and solid sections, were modeled using solid finite elements. A crack was introduced in seven different locations along the axle, resulting in one different model for each crack location, and numerical simulations of its operation were

performed. Five points along the axle were chosen as candidates for measuring and it was concluded that using data from two points, one in each axle box, is sufficient for damage detection. The presence of a crack on the axle resulted in strong harmonic components at these points, which could be clearly distinguished from data obtained for an undamaged axle. Besides having obtained good results, the axle was isolated during the simulations and further investigations on the influence of operational interference are needed; additionally, displacement data were used to assess the damage, but experimentally measuring this quantity is always more complicated.

Gómez et al. [75] proposed the use of the WPT of acceleration signals measured at the axle box of a railway bogie to detect the presence of a crack at the axle. Three accelerometers were positioned at the axle box of a real Y21 bogie (Figure 12a) and a set of tests was conducted with the operating bogie isolated on a specialized platform. During the tests, cracks with different depths were inserted into the axle (Figure 12b) to simulate the damage. A sensitivity analysis was performed for investigating the degree of correlation between the energy content of the wavelet packets and the damage severity. The most sensitive packets were chosen as suitable features to assess the damage. Based on these features, an Artificial Neural Network (ANN) was trained for automatic damage assessment. An accuracy of 96% was achieved and the rate of false alarms remained at around 5%. The same type of feature and experimental setup was used by Gómez et al. [76] to train a machine learning algorithm, more specifically the Support Vector Machine (SVM), to identify cracks on the axle. Good results were also obtained, with accuracy above 90% for cracks compromising more than 6.5% of the cross-section.

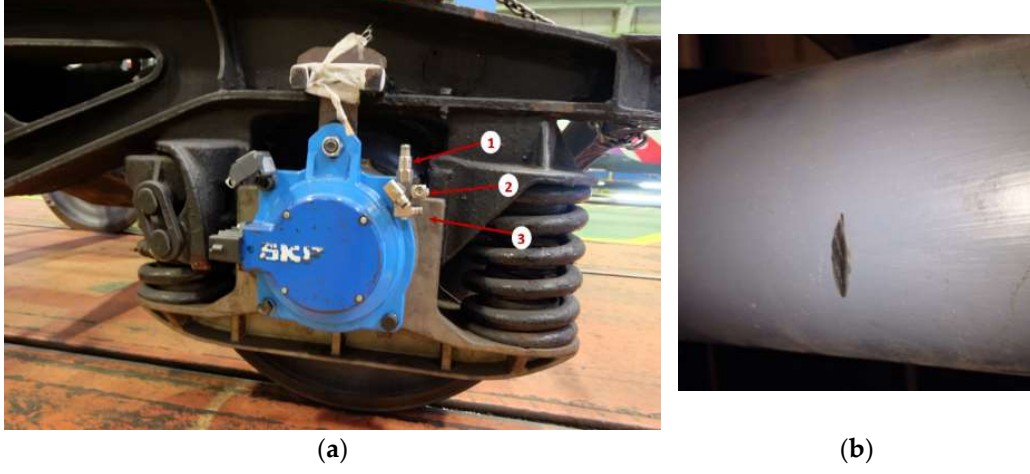

(**a**)          (**b**)

**Figure 12.** Railway axle crack identification: (**a**) experimental setup and (**b**) introduced crack [75].

Regarding wheel-related defects, Bosso et al. [77] developed an algorithm for wheel flat detection which relies on a damage indicator based on the prominence of peak acceleration in axle boxes above the baseline RMS level. Due to the cyclic nature of disturbance caused by wheel damage, the authors suggested the computation of an average acceleration signal over several wheel revolutions to avoid interference from possible track-related defects. The algorithm was validated by numerical simulations considering a track with large irregularities and also experimentally by a set of sensors installed on a freight wagon bogie (Figure 13). The method has proven to be capable of efficiently detecting flats in the early stage and inferring their severity, which is proportional to their length. Regarding the flat length, Ye et al. [78] proposed a data-driven approach for its assessment which relies on the construction of a KSM model to estimate peak axle box acceleration for a given vehicle speed and wheel flat length. The model was developed based on several numerical simulations performed with different combinations of flat length and vehicle speed. Once the model was developed a Particle Swarm Optimization (PSO) algorithm was used to obtain the flat length from measured peak axle box acceleration and vehicle speed. The methodology

was validated numerically and experimentally for an instrumented tank wagon in which a 20 mm wheel flat was artificially created. The method was able to estimate the length with a maximum error of 5% and 10% for the numerical and experimental validations, which is a very good accuracy considering all the uncertainties involved in the practical application.

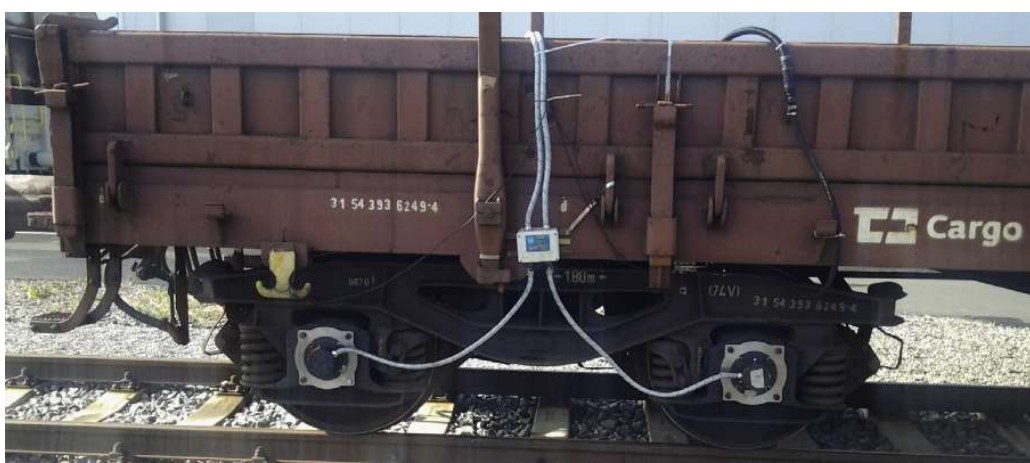

**Figure 13.** Experimental setup for wheel flat detection [77].

Differently from previous studies, which used axle box acceleration, Shi et al. [79] designed and trained a 1D lightweight Convolutional Neural Network (CNN) to detect wheel flat from car body acceleration. The proposed architecture was composed of three parts: (i) down-sampling, (ii) feature extraction, and (iii) classification. Despite being more difficult, a methodology based on accelerations in the car body may prove to be useful as it can take advantage of some instrumentation that already exists in the vehicle. The performance of the designed CNN was compared with other state-of-the-art architectures and proved to perform better with accuracies above 95% for speeds between 25 km/h and 85 km/h; however, for speeds outside this range the accuracy drops significantly. The influence of wagon and track conditions was not addressed but could have negative effects on the model accuracy.

While all the aforementioned approaches rely on digital signal processing techniques, which inevitably require processing units that consume a certain amount of energy, Bernal et al. [80] focused on the development of a fully analog wheel flat detection system with very low energy consumption, compatible with energy harvesting technologies. This approach is particularly interesting for applications involving long freight trains, in which it is not always possible to keep electrically powered acquisition and processing units in all wagons. Specifically, the system is composed of a circuit that emits a high voltage signal when acceleration reaches a certain limit, coupled with a counter to ensure that the disturbance is periodic. Finally, the device outputs only a Boolean signal indicating the presence or absence of damage. The proposed device was tested experimentally by Bernal et al. [81] using a scaled model of a railway bogie. The tests occurred for speeds of 40 km/h and 60 km/h, which are quite typical for heavy haul freight vehicles and the device was able to efficiently identify the presence of wheel flats.

Still concerning non-circular wheel wear, the assessment of wheel out-of-roundness (OOR) based on onboard instrumentation was also explored by some authors. This type of defect consists of non-uniform wear on the wheels, causing them to lose roundness and take on polygonal shapes, which is schematically represented in Figure 14. The polygonization, as depicted in the left diagram of Figure 14, can have different severities based on the amplitude of the wheel imperfections. The defect can also assume different shapes associated with the number of sides of the polygon (order); this situation is represented on the right part of the diagram in Figure 14 for the third and twentieth orders. This loss of roundness leads to excessive vibrations and, consequently, discomfort for passengers and excessive wear of vehicle and track components. In theory, the polygonization of the wheels

is responsible for generating typical frequencies as a function of vehicle speed and the order of the polygon. Based on this theory, Song et al. [82] proposed a time-frequency methodology for detecting OOR on the wheels of high-speed trains. The authors suggested the combined use of the Ensemble Empirical Mode Decomposition (EEMD) algorithm [83] and the Wigner–Ville Distribution (WVD), which is a powerful means of time–frequency signal processing for analyzing nonstationary signals. The method was tested with simulated axle box acceleration data considering track irregularities. The combined use of both techniques (EEMD and WVD) helped eliminate numerical errors and provided very accurate detection of the frequency associated with the defect. However, comprehensive experimental tests are still needed to verify the accuracy of the method against environmental effects and nonlinearities associated with suspensions and ballast behavior. Instead of using the time domain, Sun et al. [84] took advantage of the cyclic nature of wheel defects and proposed a damage feature based on the axle box acceleration as a function of the wheel rotation angle. A simple and effective alternative was adopted to eliminate the influence of non-cyclic components (not associated with wheel damage), calculating the average response for several wheel revolutions. The method was validated both numerically and experimentally and good results were achieved in detecting the damage severity and the polygon order.

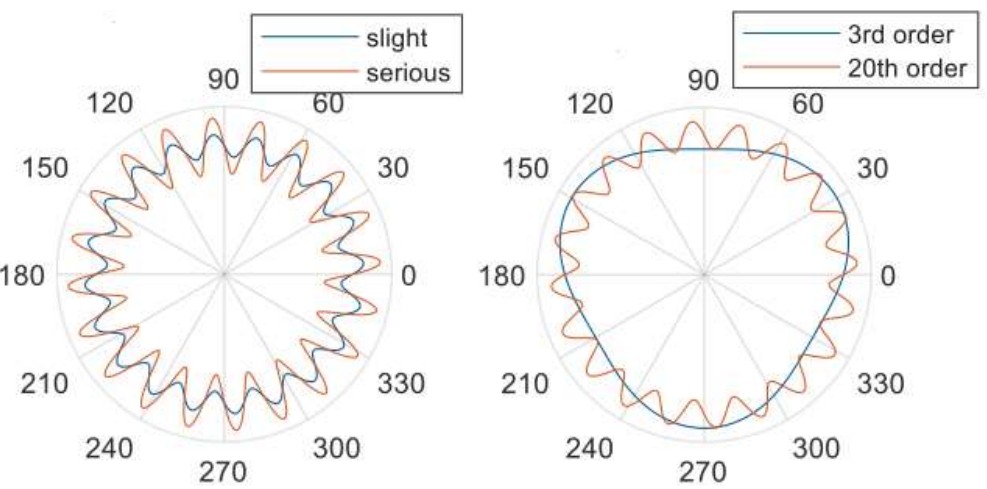

**Figure 14.** Wheel out-of-roundness (OOR) defects [84].

Remaining with the wheel OOR defect, Ye et al. [85] proposed a complete and fully automatic methodology for assessing this type of damage based on a deep learning model. The main steps of the methodology are represented in Figure 15a: firstly, wheelset acceleration time histories are measured and a feature extraction step is performed on this data. Then, a neural network-based deep learning model called OORNet was designed and trained based on a large dataset of axle box acceleration, generated through numerical simulation, comprising 2000 different wheel OOR defects, where 70% of the samples were used for training and 30% for testing the accuracy of the model. The model's architecture is schematically described in Figure 15b, comprising the operations performed in each step, as well as the size of the corresponding data arrays. As shown in the first layer, three types of features were used as inputs for the model: the upper envelope of the time domain acceleration signal, the FFT spectrum, and the time-frequency representation derived from the Wavelet transform. The features are then processed by 1DCNNs for FFT and amplitude, and 2DCNNs for the Wavelet transform. The result of this processing is then merged and processed by a Fully Convolutional Neural Network (FCNN) and a 3D CNN. Finally, the model returns two types of outputs. the predicted roughness profile of the wheel and a Boolean (faulty or healthy) indicator. based, respectively, on regression and classification strategies. During the testing phase, the model could estimate very accurately the wheel roughness profile and, as can be seen in the confusion matrix of Figure 15a, an accuracy of 97% was reached for the defect identification. The performance of the OORNet was

also compared with the other three deep learning models using different features and has proved to be superior. However, the model has been shown to be very sensitive to speed fluctuations, which is an issue needing a solution. Before the transition to practical applications, data simulated for different train speeds during the training phase may be included.

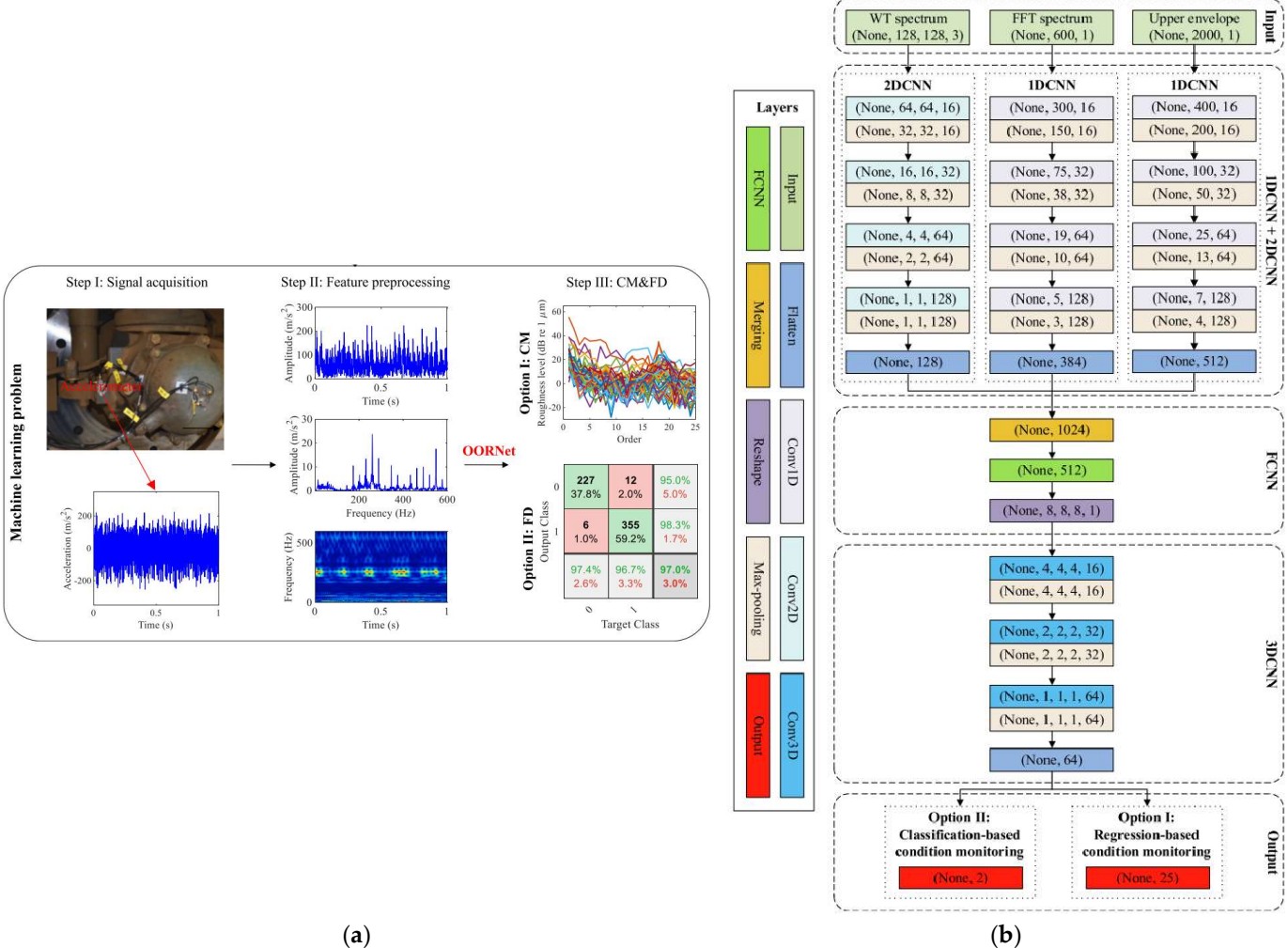

(**a**)                                                                                                                           (**b**)

**Figure 15.** Machine learning problem for wheel out-of-roundness (OOR) defect detection: (**a**) implementation framework and (**b**) architecture of OORNet model [85].

Another type of issue related to the wheels is their uneven wear in the same bogie, usually called wheel diameter difference (WDD), which is a recurring problem in wagons operating in heavy-haul lines. To monitor this condition, Xie et al. [86] proposed the use of the lateral axle box acceleration signal and developed an automatic methodology, depicted in the framework of Figure 16, to extract in-phase WDD (on the same bogie side) and anti-phase WDD (on different sides of the bogie). The feature extraction was performed based on the combined use of the Adaptive Chirp Mode Decomposition (ACMD) signal decomposition algorithm and the calculation of the Fractal Box Dimension (FBD). Then, a machine learning algorithm, the Multiple Kernel Extreme Learning Machine (MKELM), was trained for feature classification between normal operation, anti-phase WDD, and in-phase WDD. The methodology was validated based on numerical and measured data and accuracies above 95% were obtained for all cases.

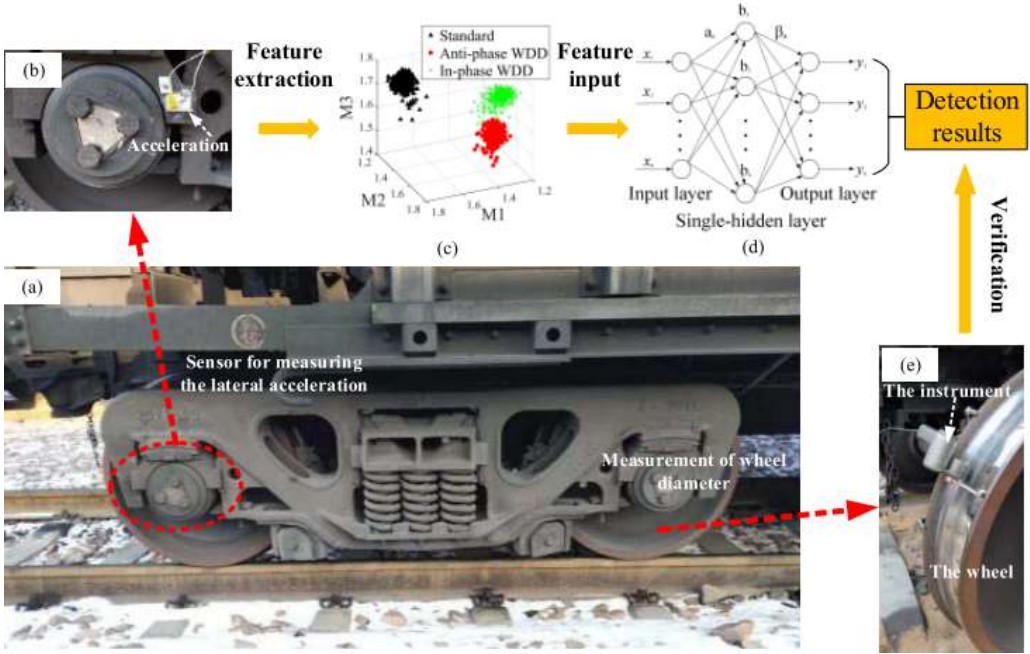

**Figure 16.** Framework of the methodology for wheel diameter difference detection: (**a**) vehicle overview, (**b**) instrumentation, (**c**) feature extraction, (**d**) classification, and (**e**) wheel diameter measurement [86].

Besides the WDD, another issue regarding the wear of train wheels is the change of the equivalent conicity profile necessary to keep the vehicle aligned to the track. Aiming to monitor this parameter based on onboard instrumentation, Kaiser et al. [87] proposed a model-based approach to identify the equivalent conicity of the wheels from the lateral acceleration of the car body and lateral acceleration and yaw rate of the bogies. An analytical formulation of the vehicles' lateral response, which among other parameters depends on the conicity, was derived and used a nonlinear state estimator, the constrained unscented Kalman filter (CUKF), to infer the conicity value from measured data. The proposed method was validated using simulated data from the commercial software SIMPACK® 18.4 and could successfully capture variations at the conicity value considering different track adhesion levels.

*4.3. Summary of Discussed Literature on Drive-by Methodologies Applied to the Vehicle Subsystem*

In Table 3, a summary of the main references discussed in the previous sections regarding the application of drive-by methodology on the railway vehicle subsystem is presented. Aspects such as the type of damage detected, study type, and main techniques used are highlighted for a better consolidation of the contents presented.

**Table 3.** Summary of the presented works dealing with drive-by methodology on the vehicle subsystem.

| References | Scope | Study Type | Methodology |
|---|---|---|---|
| Tsunashima et al. [67] | VS | N | Construction of a bank of numerical models with different failure conditions and finding, based on Kalman filters the one that best represents the measured outputs over time |
| Liu et al. [68] | VS | N/E | Use of the recursive least square adaptive filtering algorithm to |
| Wei et al. [69] | VS | N | Comparative study between some common model-based formulations and data-driven ones, namely the robust observer, the Kalman filter combined with the generalized likelihood ratio test method, the dynamical principal components analysis, and the dynamical canonical variate analysis |

**Table 3.** *Cont.*

| References | Scope | Study Type | Methodology |
|---|---|---|---|
| Mei & Ding [70] | VS | A/N | Coupling between bounce and pitch movements of bogies under suspension fault conditions |
| Dumitriu et al. [71] | VS | N/E | RMS value of bogie acceleration under normal operating conditions and for a primary suspension damper failure |
| Ye et al. [72] | VS | N/E | Multiscale algorithm permutation entropy for feature extraction is associated with a dimensionality reduction technique based on Manifold learning, particularly the local linear tangent space alignment, and a Least Squares Support Vector Machine for classification. |
| Ye et al. [73] | VS | N/E | 1D convolution neural network for feature extraction and a fully connected neural network for classification |
| Hassan et al. [74] | WS | N | Harmonics proportional to the speed of the axis in the displacement signal of the axle boxes for axle crack detection |
| Gómez et al. [75] | WS | E | Energy from Packets correlated with damage after applying a Wavelet package Transform and a neural network as a classifier for axle crack detection |
| Gómez et al. [76] | WS | E | Energy from Packets correlated with damage after applying a Wavelet package Transform and a support vector machine as a classifier for axle crack detection |
| Bosso et al. [77] | WS | N/E | The prominence of peak acceleration on axle boxes compared to the base level for wheel flat detection |
| Ye et al. [78] | WS | N/E | Kriging surrogate interpolation model correlating peak acceleration, vehicle speed, and wheel flat length for assessment of the flat length |
| Shi et al. [79] | WS | E | Convolution neural networks used for wheel flat detection from car body accelerations |
| Bernal et al. [80,81] | WS | E | An analog device equipped with accelerometers and a peak counter for identifying periodic patterns associated with wheel flat |
| Song et al. [82] | WS | N | Time-frequency representation obtained through the application of the Ensemble Empirical Mode Decomposition and Wigner-Ville algorithm Distribution for wheel out-of-roundness detection |
| Sun et al. [84] | WS | N/E | Average of several revolutions of acceleration on axle boxes plotted as a function of axle rotation angle for wheel out-of-roundness detection |
| Ye et al. [85] | WS | N | Fourier transform, amplitude of the signal envelope, and time–frequency representation derived from the Wavelet transform as inputs for a neural network-based model to detect wheel out-of-roundness |
| Xie et al. [86] | WS | N/E | Fractal Box Dimension calculated from signal decomposition by adaptive chirp mode decomposition and the multiple kernel extreme learning machine used as a classifier for wheel diameter difference detection |
| Kaiser et al. [87] | WS | N | Use of constrained unscented Kalman filter for estimating the equivalent conicity value of train wheels |

Abbreviations: Vehicle suspension (VS), wheelset (WS), analytical (A), numerical (N), and experimental (E).

## 5. Conclusions

In this article, several works regarding the development of formulations, based on data collected by onboard monitoring systems, for vehicle and track condition assessment were presented and discussed. Recent advancements in track monitoring research have led to more sophisticated methods especially for assessing track irregularities and detecting rail damage. These techniques have shown promising results in experimental validation, suggesting that they have the potential to perform the transition from research to commercial solutions. However, when it comes to detecting damage on track supporting elements, the field is still in its early stages, with most studies relying solely on numerical models based on simplified approaches. This is due to the increased complexity of these support elements and their interfaces, as well as their greater distance from the vehicle in terms of the vibration transmission trajectory. Concerning vehicle damage detection, significantly robust solutions were proposed and experimentally validated. Especially regarding the identification of wheelset damage, the methodologies appear to be well-established, with

some authors working on the development of embedded devices with good perspectives for achieving the commercial application of these technologies.

It is also relevant to point out the increased number of authors that are proposing methodologies based on artificial intelligence algorithms to identify anomalies in both track and vehicle. Inevitably, due to the long extensions of the railways and the high demand for vehicles, the feasibility of these methodologies involves the processing and interpretation of large volumes of information. Thus, from the practical point of view, making the application of these methodologies fundamentally feasible depends on developing techniques that allow an automated interpretation of the results, which is already being performed by several authors.

Despite the advances mentioned above, the performance of the methodologies developed so far, especially those involving the detection of damage to the railways, still needs to be evaluated in the face of operational disturbances. There is a lack of studies in the literature evaluating the accuracy of these methodologies, for example, in the face of measurement noise, limitations on sensor positioning, variations in vehicle operating conditions, wind interference, temperature variations, and track irregularities, among other factors. In addition, there is a lack of studies demonstrating how these methodologies can perform in scenarios where there are multiple defects involved, both on the vehicle side and on the track side. Finally, it is crucial to address the gap in research on the application of drive-by techniques in real operational scenarios. Aspects such as storage, data transmission, energy supply, onboard data processing capacity, among other aspects, need to be analyzed in more depth.

Bearing in mind the main difficulties referred to above, it is expected that in future work researchers will focus mainly on issues related to the practical applicability of drive-by monitoring through experimental validation of the methodologies proposed so far. These studies should focus on new techniques capable of isolating the influence of environmental/operational factors on damage indicators. In addition, factors such as optimal positioning and the most suitable sensors to ensure better accuracy of the methodologies will inevitably have to be investigated in detail. Concerning data processing and interpretation, further developments should bring more advanced techniques based on artificial intelligence, following the line of many recent works presented. These methodologies have proven to be increasingly powerful in extracting patterns from datasets and have great potential to increase the robustness of damage identification techniques. Finally, based on these developments, there will also be a broad field of research in the design of dedicated monitoring, processing, storage, and data transmission systems to be installed in railway vehicles, compatible with the railway's operational reality.

**Author Contributions:** Conceptualization, E.F.S., C.B. and D.R.; writing—original draft preparation, E.F.S., C.B. and A.M.; writing—review and editing, D.R., A.M. and H.C.; visualization, E.F.S. and C.B.; supervision, T.N.B. and D.R.; project administration, T.N.B. All authors have read and agreed to the published version of the manuscript.

**Funding:** The authors would like to acknowledge CNPq (Brazilian Ministry of Science and Technology Agency), CAPES (Higher Education Improvement Agency), FAPESP (São Paulo Research Foundation) for financial support under grant #2022/13045-1, VALE Catedra Under Rail, project "FERROVIA 4.0", with reference POCI-01-0247-FEDER- 046111, cofounded by the European Regional Development Fund (ERDF), through the Operational Programme for Competitiveness and Internationalization (COMPETE 2020) and the Lisbon Regional Operational Programme (LISBOA 2020), under the PORTUGAL 2020 Partnership Agreement", and project "SMART WAGONS—DEVELOPMENT OF PRODUCTION CAPACITY IN POR-TUGAL OF SMART WAGONS FOR FREIGHT", with reference C644940527-00000048, from the Incentive System to Mobilizing Agendas for Business Innovation, funded by the Recovery and Resilience Plan and by European Funds NextGeneration EU.

**Institutional Review Board Statement:** Not applicable.

**Informed Consent Statement:** Not applicable.

**Data Availability Statement:** No new data were created or analyzed in this study. Data sharing is not applicable to this article.

**Conflicts of Interest:** The authors declare no conflict of interest.

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
