# Peer review of "Drive-by Methodologies Applied to Railway Infrastructure Subsystems: A Literature Review—Part II: Track and Vehicle"

_applsci, doi:10.3390/app13126982_

Round 1

Reviewer 1 Report

this paper focuses on the indirect identification and evaluation of railway vehicle and track, also the bridge subsystem is reviewed. The state-of-the-art of drive-by monitoring is well illustrated. The paper is very important to the develpment of theory and application in railway engineering because it is more possible to be used than road field. 

(1) some reference number is missed, which induces mistake in the paper, at least two times such as page 2.

(2) some papers are not very clear, the quality can be improved.

this paper focuses on the indirect identification and evaluation of railway vehicle and track, also the bridge subsystem is reviewed. The state-of-the-art of drive-by monitoring is well illustrated. The paper is very important to the develpment of theory and application in railway engineering because it is more possible to be used than road field. 

(1) some reference number is missed, which induces mistake in the paper, at least two times such as page 2.

(2) some papers are not very clear, the quality can be improved.

Reviewer 2 Report

In the second part of the article Drive-by methodologies applied to railway infrastructure sub-systems, the authors also demonstrated a highly scientific approach.

Part II: Track and vehicle, very clearly shows the data acquisition methods that are very important for structural analysis.

Even in this part, after the introduction, I am missing a subsection on the purpose of the research and limitations of the research.

In Chapter 2, the “Error! Reference source not found", which prevents the reviewer from viewing the cited work.

Fix text size in figure 10 and 16.

I ask the authors to describe in more detail figures 22 and 23

I miss the discussion before the conclusion, where the authors should summarize the essence of the research and the results

The conclusion is well written and describes the problems of the research.

Reviewer 3 Report

This is interesting and timely work. Some comments:

1.       The same as the first part, the abstract is quite general. Please be more specific, for example list several future practical applications.

2.       What are the differences between train embedded sensors and vehicle embedded sensors, and what are the differences between railway and roads?

3.       Introduction: according to my knowledge, there are a lot of efforts from China in ‘drive-by methodologies for condition assessment of railway bridges’. Please revise it.

4.       Table 1: the literatures are quite limited, please double check.

5.       Figure 1 is perfect.

6.       Figure 23b can be rotated, it is not easy to understand now.

7.       The future perspective is reliable. 

No
